# Design of Robust Sparse Wideband Beamformers with Circular-Model Mismatches Based on Reweighted $\ell_{2,1}$ Optimization

Yu Bao [1,*] , Haixiao Zhang [1], Xiaoli Liu [1], Yuhan Jiang [1] and Yu Tao [2,3]

1 College of Electronic and Information Engineering, Changzhou Institute of Technology, Changzhou 213031, China; zhanghax@czu.cn (H.Z.)
2 College of Electronic and Information Engineering, Nanjing University of Aeronautics and Astronautics, Nanjing 211100, China; taoyu@cslg.edu.cn
3 School of Electronic and Information Engineering, Changshu Institute of Technology, Suzhou 215500, China
* Correspondence: baoyu@czu.cn

**Abstract:** Wideband beamformers have been widely studied in wireless communication, remote sensing and so on. Generally speaking, to improve the spatial filtering ability of beamformers, there usually needs more sensors, which implies increased computational complexity and hardware costs. Besides that, wideband beamformers are known to be exceedingly sensitive to sensor mismatches in practice. Nevertheless, there is still a gap in research on the design of robust sparse wideband beamformers. In this paper, a two-step design of this topic is proposed. Firstly, a robust design based on the worst-case performance optimization (WCPO) using circular-model (CM) sensor mismatches is reformulated to address shortcomings of constraint sensitivity. Secondly, inspired by the joint sparse technology in compressive sensing theory, we focus on the sparse design of wideband beamformer. The constraints for the response characteristics and robustness are set from first step, and an iterative algorithm based on reweighted $\ell_{2,1}$ optimization is adopted to achieve maximum sparsity of the sensor array. The mainly advantages of the work are that the proposed design exhibits accordant performance in terms of response and robustness, but few sensors compared with the counterpart with uniform array. Moreover, we surprisingly find that the optimized sparse array is also applicable to other design based on WCPO criterion. Simulation results are provided to verify the superior of the proposed methods compared to the existing counterparts.

**Keywords:** wideband beamformers; sensor arrays; worst-case performance optimization; robustness; sparse array

## 1. Introduction

Beamforming, as one of the key technologies in array signal processing, has widely applied in practice, such as wireless communication, medical imaging, radar, remote sensing, etc. [1–12]. In wireless communication and remote sensing array signal processing fields, wideband signals are mostly concerned and wideband beamformers based on finite impulse response (FIR) of sensor arrays have been applied in recent years. Wideband beamformer in general can be categorized as data-dependent beamformer or data-independent beamformer [1]. By contrast, data-independent beamformers, also called fixed beamformers, have benefits of low computational complexity and the offline preset beamformer weights. Moreover, they do not require to consider the interference of desired signal coherence [13]. Accordingly, data-independent wideband beamformers provide appropriate degrees of design freedom and related works have been extensively studied [14–18].

It is conventional that the beamforming performance can be improved by increasing the number of sensors in a uniform linear array (ULA) [1]. Unfortunately, this leads to more computational complexity and higher hardware costs. One desirable alternative

is the sparse arrays, which can provide more degrees of freedom to achieve a better beamforming performance with nonuniform array or less number of sensors in contrast to the uniform one [19–22]. Some efforts have been made to optimize the location of sensors for sparse array, such as Genetic Algorithm [19,20], Simulated Annealing [21], Matrix Pencil Method [22], to reach an acceptable performance for array response. However, these nonlinear optimizations suffer potential extended computing time and non-optimal results. It is noticed the sparse estimation methods for direction of arrival using compressive sensing (CS) methodology have been widely applied [23–27]. Motivated by techniques in CS fields, several sparse array designs have been proposed in narrowband beamforming [28,29]. In contrast to narrowband, sparse array optimizations for wideband require sparse all FIR tap coefficients along the associated sensor, resulting in a lack of in-depth studies. Recently, a design of wideband sparse array based on complex-valued norm minimization has been proposed in [30], successfully achieving a frequency invariant beamformer with sparse array. Although this work achieves sparse design and shows efficiency in reaching the optimal solution compared to GA design, there is still a gap that whether the CS-based method can achieve benefits in terms of hardware or computation costs compared to conventional ULAs with same aperture size.

Another problem of interest is the robustness of sparse array. As well known, the characteristics of sensors are usually influenced by sensor mismatches, i.e., sensor gain, phase and location errors, which are normally inevitable in applications [13]. Conventional wideband beamformers are sensitive to these mismatches, especially for super-directive beamforming [15,17,18]. To combat the problem, it is necessary to design robust wideband beamformers. The classic approaches utilize white noise gain (WNG) constraints to achieve robustness [31]. As the WNG is not directly related to sensor errors, it is challenging to ensure the optimal robustness [15]. Over the past decade, many efforts have been made to improve robust designs by using some prior knowledge of sensor mismatches [13,15,32–38]. Among them, there is one type of approaches that utilizes unknown but bounded mismatches on sensor characteristics using the worst-case performance optimization (WCPO) criterion. Representative designs include the second-order cone programming (SOCP) based design approach [32], the semidefinite programming (SDP) based approach [38] and the design using a circular-model (CM) variation to represent the sensor mismatches [36]. However, it is worth noting that, to the best of our knowledge, the research on the design of robust sparse wideband beamformers is still left blank.

Recall that the CM-based approach adopts a two-phase design, which achieves robust design using WCPO criterion in the first step and further improves the WNG in the second step [36] . In this paper, we refer to the two-phase strategy of CM-based method to fulfill the design of robust sparse wideband beamformers. To ensure a more economical sparse beamformer than existing ULA one, there needs a wideband beamformer with good performance and robustness as a reference. Consequently, we attempt to remedy the problems of existing CM-based method. Although the CM-based design has been demonstrated to be superior to SOCP-based counterpart, the performance of array response may degrade significantly with tight stopband level constraint [38]. To address the problem, a reformulate CM (RCM) representation of mismatches is proposed by applying the first-order Taylor series expansion, and an efficient rectangular region of variation of sensor errors is expressed. To proceed, we separate the cost function into the part of passband and the part of robustness to efficiently adjust the tradeoff between beamforming passband performance and robustness. Afterwards, we propose a two-phase robust sparse design method. In the first phase, the improved RCM-based method with ULA is used to derive the response performance and robustness constraints. In the second phase, an iterative algorithm based on reweighted $\ell_{2,1}$ optimization is made to reduce the number of sensor elements and maintain the similar beamforming performance as the counterpart in the first phase. Inspired by joint sparsity of the multiple measurement vectors [24–26], we divide the aperture into dense discrete grids, and the sensor locations are assumed to lie on a gird. As a consequence, the coefficients of a beamformer can be represented as a row-sparse

matrix, where each row corresponds to a gird and the elements of a row are FIR taps, and the sparsity of a row-sparse matrix can be defined as the number of nonzero rows. In order to achieve further sparse solution, we herein introduce an iterative strategy to construct the weighting terms for better estimation of the nonzero coefficient locations, which takes advantage from a reweighted $\ell_1$-norm minimization [27,28]. Experiment results show that the sparse design provides less sensors than the ULA, but achieves similar beamforming performance, filling the gap in sparse array beamforming design. Moreover, it is also interesting noticed that the proposed optimal sparse array can be applied in different WCPO-based robust wideband beamformers and meets the above results, which provides an insight into the theoretical support for practical applications.

As for the rest of the paper, it is organized as follows. Section 2 describes the mathematical model of wideband beamformers with sensor mismatch and an introduction to existing related methods. In Section 3, we present the proposed RCM-based design and a sparse array design method based on reweighted $\ell_{2,1}$ norm optimization is further derived. Section 4 presents several simulation examples and comparison results to illustrate the advantages of the proposed design. Section 5 concludes the paper.

## 2. Background

### 2.1. Problem Statement

Consider a filter-and-sum structured wideband beamformer with $M$ sensors in farfield, followed by an $L$-tap FIR filter for each sensor, as shown in Figure 1. Without losing generality, the center of the array is set to the reference point. Define the coefficients of FIR filter as $\mathbf{w}_m = [w_{m,0}, w_{m,1}, \ldots, w_{m,L-1}]^T \in \mathbb{R}^{L \times 1}$, where $w_{m,l}$ is the $l$th factor of the $m$th sensor and $(\cdot)^T$ represents the transpose. Therefore, the weight vector of wideband beamformer is given by

$$\mathbf{w} = \left[ \mathbf{w}_0^T, \mathbf{w}_1^T, \cdots, \mathbf{w}_{M-1}^T \right] \in \mathbb{R}^{ML \times 1}. \tag{1}$$

As a result, the array response of the wideband beamformer with the angle $\vartheta$ at frequency $f$ can be expressed as

$$B(\vartheta, f) = \mathbf{w}^T \mathbf{g}(\vartheta; f) \tag{2}$$

where $\mathbf{g}(\vartheta; f) = \mathbf{h}(\vartheta; f) \otimes \mathbf{e}(f)$ denotes the vector of array response, $\otimes$ is the Kronecker product, and we have

$$\mathbf{h}(\vartheta; f) = [h_0(\vartheta; f), h_1(\vartheta; f), \cdots, h_{M-1}(\vartheta; f)]^T \tag{3}$$

$$\mathbf{e}(f) = \left[ 1, e^{-j2\pi f/f_s}, \cdots, e^{-j2\pi f(L-1)/f_s} \right]^T \tag{4}$$

with $f_s$ being the sampling frequency, and $j = \sqrt{-1}$ in (4). The sensor transfer function $h_m(\vartheta, f)$ in (3) is given by

$$h_m(\vartheta; f) = \exp(-j2\pi f d_m \cos \vartheta / c) \tag{5}$$

where $d_m$ expresses the distance between the reference point and the $m$th sensor, and $c$ represents the speed of signals. For ease of description, we also define the following vectors:

$$\mathbf{g}(\vartheta; f) = [\mathbf{g}_0(\vartheta; f)^T, \mathbf{g}_1(\vartheta; f)^T, \ldots, \mathbf{g}_{M-1}(\vartheta; f)^T]^T \tag{6}$$

with $\mathbf{g}_m(\vartheta; f) = [g_{m,0}(\vartheta; f), g_{m,1}(\vartheta; f), \ldots, g_{m,L-1}(\vartheta; f)]^T$.

When existing sensor mismatches, the $m$th sensor characteristics can be determined by

$$\kappa_m(\vartheta; f) = [1 + \Delta a_m(\vartheta; f)] \exp[-j\Delta \phi_m(\vartheta; f)] \tag{7}$$

where $\Delta\phi_m(\vartheta; f) = \Delta\varphi_m(\vartheta; f) + 2\pi f \Delta d_m \cos\vartheta/c$, $\Delta a_m(\vartheta; f)$, $\Delta\varphi_m(\vartheta; f)$, and $\Delta d_m$ refer to the $m$th sensor gain, phase and location errors, respectively. According to the WCPO criterion, $\Delta a_m(\vartheta; f)$, $\Delta\varphi_m(\vartheta; f)$ and $\Delta d_m$ are assumed to be random, independent but bounded. Specifically, assign $|\Delta a_m| \leq \delta_a$, $|\Delta\varphi_m| \leq \delta_\varphi$, and $|\Delta d_m| \leq \delta_d$ , where $\delta_a$, $\delta_\varphi$ and $\delta_d$ are the disturbance boundaries of sensor errors. For ease of notation, $(\vartheta; f)$ of errors will be omitted. As a result, the sensor transfer function with sensor mismatches can be derived as

$$\tilde{h}_m(\vartheta; f) = \kappa_m(\vartheta; f)\exp(-j2\pi f d_m \cos\vartheta/c) \tag{8}$$

Thus, the array response with sensor mismatches is given by

$$\hat{B}(\vartheta; f) = \mathbf{w}^T \hat{\mathbf{g}}(\vartheta; f) \tag{9}$$

where $\hat{\mathbf{g}}(\vartheta; f) = \hat{\mathbf{h}}(\vartheta; f) \otimes \mathbf{e}(f)$, and $\hat{\mathbf{h}}(\vartheta; f) = \left[\tilde{h}_0(\vartheta; f), \tilde{h}_1(\vartheta; f), \cdots, \tilde{h_{M-1}}(\vartheta; f)\right]^T$

The design problem is to find an optimum weight vector $\mathbf{w}$ under certain design criterion in order that the array response with mismatches $\hat{B}(\vartheta; f)$ excellently matches a preset $B_d(\vartheta; f)$, given the description of unknown but bounded sensor mismatches.

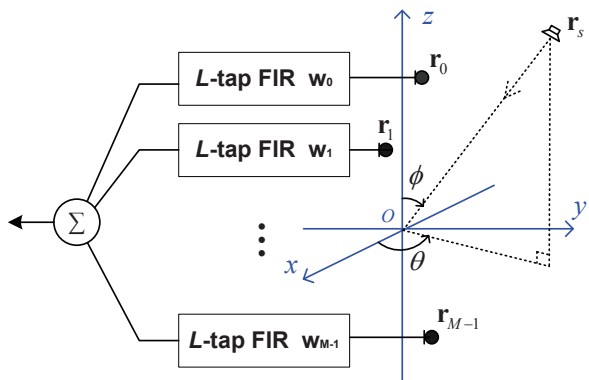

**Figure 1.** Configuration of wideband beamformer with filter-and-sum structure.

### 2.2. Existing CM-Based Design Approach

Herein, we briefly introduce a robust design approach based on a CM representation of sensor mismatches [36]. Define $\vartheta_p$, $\vartheta_s$ and $f_i$ are discrete points in passband, stopband and frequency band, respectively. The optimization problem for CM-based design is derived as

$$\min_{\mathbf{w}} \quad \max_{\vartheta_p, f_i}\left\{\left|\mathbf{u}^T\mathbf{q}(\vartheta_p, f_i)\mathbf{w}^T\mathbf{g}(\vartheta_p, f_i) - B_d(\vartheta_p, f_i)\right| + r_\chi \sum_{m=0}^{M-1}\left|\mathbf{w}_m^T\mathbf{g}_m(\vartheta_p, f_i)\right|\right\} \tag{10}$$

$$\text{s.t.} \quad \max_{\vartheta_s, f_i}\left\{\left|\mathbf{u}^T\mathbf{q}(\vartheta_s, f_i)\mathbf{w}^T\mathbf{g}(\vartheta_s, f_i)\right| + r_\chi \sum_{m=0}^{M-1}\left|\mathbf{w}_m^T\mathbf{g}_m(\vartheta_s, f_i)\right|\right\} \leq \Gamma_{sb}.$$

where $\mathbf{u} = [1, j]^T$ for complex calculations. $\mathbf{q}(\vartheta; f) \in \mathbb{R}^{2\times1}$ and $r_\chi$ represent the center and the radius of the circle in the complex plane, respectively. $\Gamma_{sb}$ is a parameter that controls the stopband level and $B_d(\vartheta_p, f_i)$ is the preset passband response.

### 3. Proposed Design Approaches

In this section, we firstly provide a simplified variation of sensor mismatches, and then reformulate the cost function of the CM-based design into several parts to overcome the shortcomings of the CM-based approach. In addition, an iterative algorithm based on reweighted $\ell_{2,1}$ optimization is proposed to achieve the sparsity maintaining robustness simultaneously.

### 3.1. Reformulation Approach

Consider the condition with existing minor sensor mismatches, by applying ternary first-order Taylor series expansion to (7) at the point $(\Delta a_m = 0, \Delta \varphi_m = 0, \Delta d_m = 0)$, the $m$th sensor characteristics can be approximated as [34]

$$\kappa_m(\vartheta; f) \simeq 1 + \Delta a_m - j(\Delta \phi_m). \tag{11}$$

Consequently, the real and imaginary parts of $\kappa_m(\vartheta; f)$ are respectively given by

$$\text{Re}\{\kappa_m(\vartheta; f)\} = 1 + \Delta a_m \tag{12}$$
$$\text{Im}\{\kappa_m(\vartheta; f)\} = -\Delta \phi_m. \tag{13}$$

Since all sensor errors are bounded, the variation of sensor errors can be actually bounded to the rectangular area, which is marked in gray color in Figure 2. It can be derived from the geometric that the center of the gray area is at $(1, 0)$ and the lengths of the two sides are $2\delta_a$ and $2[\delta_\varphi + |2\pi f \delta_d \cos \vartheta / c|]$ respectively. One key feature of the CM-based design is to define a compact set of circle to encompass the sensor errors variation. Evidently, the minimum-area circle is the tangent circle represented by the dashed line in Figure 2, and the circle center is coincidentally located at $(1, 0)$, and the radius is

$$r'_\chi = \sqrt{\delta_a^2 + (\delta_\varphi + |2\pi f \delta_d \cos \vartheta / c|)^2}. \tag{14}$$

By (14), the CM-based approach can be simplified as

$$\min_{\mathbf{w}} \quad \max_{\vartheta_p, f_i} \left\{ |\mathbf{w}^T \mathbf{g}(\vartheta_p, f_i) - B_d(\vartheta_p, f_i)| + r'_\chi \sum_{m=0}^{M-1} \left| \mathbf{w}_m^T \mathbf{g}_m(\vartheta_p, f_i) \right| \right\} \tag{15}$$

$$\text{s.t.} \quad \max_{\vartheta_s, f_i} \left\{ |\mathbf{w}^T \mathbf{g}(\vartheta_s, f_i)| + r'_\chi \sum_{m=0}^{M-1} \left| \mathbf{w}_m^T \mathbf{g}_m(\vartheta_s, f_i) \right| \right\} \leq \Gamma_{sb}.$$

Note that the drawback of origin CM-based design is that the passband performance is sensitive to the setting parameters, and may degrade badly if the constraints are chosen tightly. To deal with this problem and earn the passband performance, we split the cost function into two parts: the passband response part and the robustness part. Therefore, the optimization problem can be further reformulated as the following expression:

$$\min_{\eta, \mathbf{w}} \eta \tag{16}$$

$$\text{s.t.} \begin{cases} |\mathbf{w}^T \mathbf{g}(\vartheta_p, f_i) - B_d(\vartheta_p, f_i)| \leq \lambda, \\ r'_\chi \sum\limits_{m=0}^{M-1} |\mathbf{w}_m^T \mathbf{g}_m(\vartheta_p, f_i)| \leq \eta - \lambda, \\ |\mathbf{w}^T \mathbf{g}(\vartheta_s, f_i)| + r'_\chi \sum\limits_{m=0}^{M-1} |\mathbf{w}_m^T \mathbf{g}_m(\vartheta_s, f_i)| \leq \Gamma_{sb}. \end{cases}$$

Through such reformulation, the passband performance and robustness of the beamformer are independently considered and constrained, thus having the potential to better control the trade-off between the two. This will be verified in subsequent experiments. Besides, the cost function and constraints in (16) are all convex, thus the convex optimization packages such as CVX [39] can efficiently solve (16).

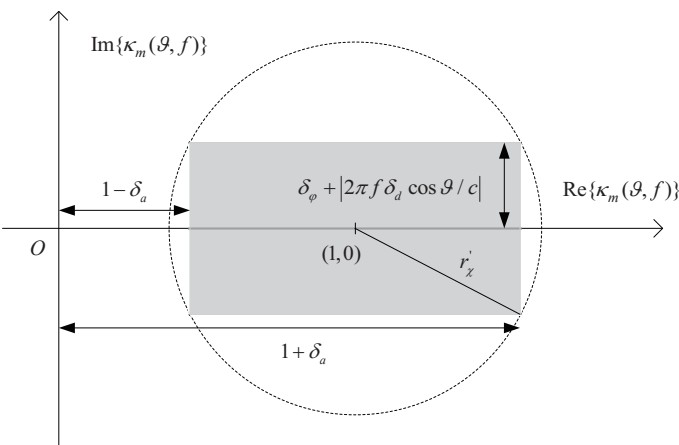

**Figure 2.** Exhibition for the variation of $\kappa_a(\vartheta; f)$. The gray rectangular is the region of sensor mismatches. The dashed circle is the tangent circle of the rectangular.

### 3.2. Sparse Design via Reweighted $\ell_{2,1}$ Optimization

In general, beamformers can reach better performance by expending the number of the sensors, resulting in an increased cost. Inspired by CS theory, the sparse design of narrowband beamforming has been studied, the array response can fit the desired response or an acceptable performance level with minimally sensors [28,29]. Nevertheless, different with the narrowband, each sensor of wideband beamformers array has a FIR filter behind. Reducing the sensor should minimize all filter weights to zero simultaneously and then the corresponding sensor can be considered to be inactive and eliminated to fulfill the sparse design, e.g., $\mathbf{w}_m = \mathbf{0}$ [30]. Recall the design idea of narrowband sparse array based on CS representation, all potential locations of the sensors lie on a prescribed uniform grid of the array aperture size, and the grid needs to be dense enough to ensure the optimum results are close to the grid points, which reminds a large enough $M$. Sparse beamformers are optimized to match the desired response by selecting as few non-zero valued weights as possible. In the case of wideband, we continue to use the discrete grid for array aperture. The main difficulty is how to minimize plenty of $\mathbf{w}_m$ to zeros simultaneously. To achieve this, we firstly introduce the $l_{2,0}$ norm and $l_{2,1}$ norm [26].

Suppose $\mathbf{w} \in \mathbb{R}^{ML \times 1}$ and $\mathbf{w}_m$ are defined same as in (1) above.

**Property 1.** *the $\ell_{2,0}$ norm of $\mathbf{w}$ is given by:*

$$\|\mathbf{w}\|_{2,0} = \#\{m : \|\mathbf{w}_m\|_2 > 0\} \tag{17}$$

*counts the nonzero entries of $\mathbf{w}_m$.*

**Property 2.** *the $\ell_{2,1}$ norm of $\mathbf{w}$ is expressed as:*

$$\|\mathbf{w}\|_{2,1} = \sum_{m=0}^{M-1} \|\mathbf{w}_m\|_2 \tag{18}$$

With Property 1, it is clear that the sparse design can achieve the minimization of sensors by minimizing the $\ell_{2,0}$ norm of $\mathbf{w}$ and penalize all non-zero valued $\mathbf{w}_m$. However, the optimization problem using $\ell_{2,0}$ is NP hard to solve. One of efficient methods is convex relaxation, and the tightest convex relaxation of $\ell_{2,0}$ norm is given by the $\ell_{2,1}$ norm defined

as in Property 2. Therefore, the design of minimize sensor number of wideband beamformer problem can be derived as

$$\min_{\mathbf{w}} \quad \|\mathbf{w}\|_{2,1} \tag{19}$$

$$\text{s.t.} \begin{cases} \max_{\vartheta_p, f_i}\left\{|\mathbf{w}^T\mathbf{g}(\vartheta_p, f_i) - B_d(\vartheta_p, f_i)|\right\} \leq \Gamma_{pb}^{ph1} \\ \max_{\vartheta_s, f_i}\left\{|\mathbf{w}^T\mathbf{g}(\vartheta_s, f_i)|\right\} \leq \Gamma_{sb}^{ph1} \\ \max_{f_i} \|\mathbf{w}^T\mathbf{\Lambda}(f_i)\|_2 \leq \Gamma_{G}^{ph1} \end{cases}$$

where $\mathbf{\Lambda}(f) = \mathbf{I}_M \otimes \mathbf{e}(f)$, $\mathbf{I}_M$ is a $M$-dimensional unit matrix. $\Gamma_{pb}^{ph1}$, $\Gamma_{sb}^{ph1}$ and $\Gamma_{G}^{ph1}$ are the resulting passband ripple, stopband attenuation and WNG of the proposed method in (16) and given by

$$\Gamma_{pb}^{ph1} = \max_{\vartheta_p, f_i}\left\{|\mathbf{w}_{op1}^T\mathbf{g}(\vartheta_p, f_i) - B_d(\vartheta_p, f_i)|\right\}, \tag{20}$$

$$\Gamma_{sb}^{ph1} = \max_{\vartheta_p, f_i}\left\{|\mathbf{w}_{op1}^T\mathbf{g}(\vartheta_s, f_i)|\right\}, \tag{21}$$

$$\Gamma_{G}^{ph1} = \max_{f_i} \|\mathbf{w}_{op1}^T\mathbf{\Lambda}(f_i)\|_2, \tag{22}$$

and $\mathbf{w}_{op1}$ is the optimized wideband beamformer weight vector in (16).

The sparse design approach (19) is a convex relaxed problem, which leads to part of optimized $\mathbf{w}_m$ that can be thinned out are only closed to zeros. When the optimization problem (19) is solved, $\mathbf{w}_m$ should be set to zeros while $\|\mathbf{w}_m\|_2$ is less than a user-set threshold, and achieves the design of wideband beamformers with sparse array. However, this $\ell_{2,1}$ minimization suffers that there may be plenty of directly adjacent grids, which makes it difficult to accurately determine the active sensor locations.

To further improve the sparse results and obtain better approximation to the results of $\ell_{2,0}$ norm, we firstly recall the iterative reweigthed $\ell_1$ norm minimization employed in the narrowband sparse arrays [28]. Therefore, we refer to reweighting all $\|\mathbf{w}_m\|_2$ as large weights used to discourage active locations and minor weights used to encourage others. Thus, the reweighted sparse design based on $\ell_{2,1}$ norm minimization can be expressed as

$$\min_{\mathbf{w}} \quad \sum_{m=0}^{M-1} a_m^{(i)}\|\mathbf{w}_m^{(i)}\|_2 \tag{23}$$

$$\text{s.t.} \begin{cases} \max_{\vartheta_p, f_i}\left\{|\mathbf{w}^{(i)T}\mathbf{g}(\vartheta_p, f_i) - B_d(\vartheta_p, f_i)|\right\} \leq \Gamma_{pb}^{ph1} \\ \max_{\vartheta_s, f_i}\left\{|\mathbf{w}^{(i)T}\mathbf{g}(\vartheta_s, f_i)|\right\} \leq \Gamma_{sb}^{ph1} \\ \max_{f_i} \|\mathbf{w}^{(i)T}\mathbf{\Lambda}(f_i)\|_2 \leq \Gamma_{G}^{ph1} \end{cases}$$

where

$$a_m^{(i)} = \left(\|\mathbf{w}_m^{(i-1)}\|_2 + \epsilon\right)^{-1} \tag{24}$$

is the reweighting term and $i$ is the iteration index. The parameter $\epsilon$ is a constant that provides stability and ensures the zero-valued components not strictly prohibiting a nonzero estimate at the next iteration. Apparently, in each iteration $a_m^{(i)}$ becomes large with small non-zero or nearly-zero valued coefficients, which would keep the coefficients closed to zeros in the next step, and turns out to small with large coefficients. Therefore, the problem penalizes all non-zero valued coefficients uniformly, leading to a better approximation to $\ell_{2,0}$ norm minimization. Specifically, if we set $i = 0$ and $a_m = 1$ for $m = 0, 1, \ldots, M-1$, and run iteration only once, the optimization problem (23) can be simplified to the one (19).

It is worth noting that although the amounts of coefficients in optimized **w** after the iteration are extremely close to zero but not equal to zero. We need to choose an appropriate threshold value, i.e., $\rho$, to zero these small values. In addition, the termination condition can be set for a specific number of iterations or expected sparse array sensors, depending on design requirements. On the whole, the flow diagram of iterative algorithm that alternates between estimating non-zero beamforming weights and redefining the reweighting terms is shown in Figure 3. The specific input and initial parameters and steps are also provided in Algorithm 1 as follows

---

**Algorithm 1** The Iterative Reweighted Algorithm based on $\ell_{2,1}$ norm minimization

---

**Input:** Initial weight vector: The optimized results $\mathbf{w}^{(0)}$ by (19); The resulting solution in (16): $\Gamma_{pb}^{ph1}$, $\Gamma_{sb}^{ph1}$ and $\Gamma_G^{ph1}$; Maximum iterations: $i_d$; Number of sensors used in (16): $M_{ULA}$; Threshold value: $\rho$

**Output:** optimal $\mathbf{w}^{(i)}$

1: initial $i = 1$;
2: **repeat**
3:     Calculate $a_m^{(i)}$ by (24);
4:     Solve the reweighted $\ell_{2,1}$ norm minimization (23);
5:     $i = i + 1$;
6: **until** $i = i_d$ or active number of sensors $M_s < M_{ULA}$;
7: Set $\mathbf{w}_m^{(i)} = \mathbf{0}$ when $\|\mathbf{w}_m^{(i)}\| \leq \rho$.

---

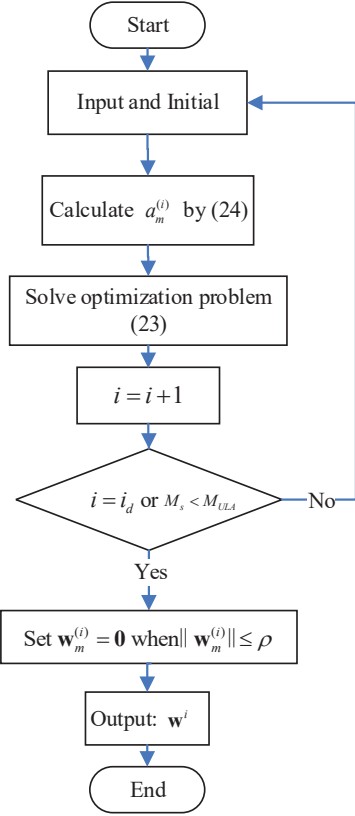

**Figure 3.** The flow diagram of the reweighted sparse design based on $\ell_{2,1}$ norm minimization.

As mentioned above, the convex minimization (19) and (23) can be optimized by the CVX package as well.

## 4. Numerical Results

In this section, we show several design examples to elucidate the performance of the proposed RCM-based and Reweighted $\ell_{2,1}$-based design approaches. For ease of comparison with design approaches, we first introduce some performance measures.

For the passband performance, the maximum change in the array response magnitude of called *Passband Ripple* is defined as

$$\sigma_p = 20 \log_{10} \left\{ \frac{\max\limits_{\vartheta_p, f_i} |B(\vartheta_p, f_i)|}{\min\limits_{\vartheta_p, f_i} |B(\vartheta_p, f_i)|} \right\}. \tag{25}$$

Moreover, beamformers with passband frequency invariance are usually provided to maintain the undistorted passband signals. To study this performance, the *Passband Invariance Factor (PIF)* is given by [18]

$$\text{PIF} = 10 \log_{10} \left\{ \frac{\frac{1}{K} \sum_{k=1}^{K} \sum_{n=1}^{N_p} \left[ \left| B(\vartheta_p^{(n)}, f^{(k)}) \right| - \mathcal{M}(\vartheta_p^{(n)}) \right]^2}{\sum_{n=1}^{N_p} \mathcal{M}^2(\vartheta_p^{(n)})} \right\} \tag{26}$$

where

$$\mathcal{M}(\vartheta_p^{(n)}) = \frac{1}{K} \sum_{k=1}^{K} \left| B(\vartheta_p^{(n)}, f^{(k)}) \right| \tag{27}$$

It is noted that a larger PIF implies worse frequency-invariant characteristic.

For the stopband performance, the maximum value of the array response magnitude named as *Stopband Level* is given by

$$\sigma_s = 20 \log_{10} \left\{ \frac{\max\limits_{\vartheta_p, f_i} |B(\vartheta_p, f_i)|}{\max\limits_{\vartheta_s, f_i} |B(\vartheta_s, f_i)|} \right\}. \tag{28}$$

For the robustness of the beamformes, we take sensor mismatches into account and define *worst passband ripple* (WPR) and *worst stopband attenuation* (WSA) respectively as

$$\sigma_p^{(wc)} = \max_{n=1,\cdots,\mathcal{N}} 20 \log_{10} \left\{ \frac{\max\limits_{\vartheta_p, f_i} |\hat{B}(\vartheta_p, f_i)|}{\min\limits_{\vartheta_p, f_i} |\hat{B}(\vartheta_p, f_i)|} \right\} \tag{29}$$

$$\sigma_s^{(wc)} = \min_{n=1,\cdots,\mathcal{N}} 20 \log_{10} \left\{ \frac{\max\limits_{\vartheta_p, f_i} |\hat{B}(\vartheta_p, f_i)|}{\max\limits_{\vartheta_s, f_i} |\hat{B}(\vartheta_s, f_i)|} \right\} \tag{30}$$

where $\hat{B}(\vartheta_p, f_i)$ and $\hat{B}(\vartheta_s, f_i)$ are the actual passband response and stopband response at the $n$th test. In the following examples, each results is obtained from $\mathcal{N} = 100$ tests.

In the simulations, a ULA with $M = 7$ is considered, and the inter-element distance $d_m = 4$ cm, which means the array aperture is 0.24 m. Set the FIR tap length $L = 20$, and the frequency band $[1500, 3500]$ Hz. $f_s$ is set to 8000 Hz, and the speed of signal is specified as sound in air, i.e., $c = 340$ m/s. The passband and stopband range are chosen as $[80°–100°]$ and $[0°–60°] \cup [120°–180°]$, respectively. The desired $B_d(\vartheta; f)$ is prescribed as $\exp\{-j2\pi f(L-1)f_s^{-1}/2\}$, conforming to group-delay. The sensor gain, phase, and location errors are supposed to follow uniform distribution in $[-0.05, 0.05]$, $[-5°, 5°]$, and $[-0.001, 0.001]$ m, respectively. The beampatterns presented in this study are obtained by averaging the results of 100 Monte Carlo trials, each involving random samples of sensor mismatches. In order to facilitate comparison, the beamformer weights are normalized such

that the maximum of the mean array response is equal to unity. The wildly used convex optimization package CVX [39] is employed to solve the associated design problems.

### 4.1. Example 1 and 2

In this subsection, we compare the CM-based and the proposed RCM-based design in Section 3.1. In the first example, we set the parament of stopband level $\Gamma_{sb} = -6$ dB, and the beampatterns of the wideband beamformers designed by the two approaches are shown in Figure 4a,b. The second example shows the resultants with a tighter stopband level constraint of $\Gamma_{sb} = -13$ dB, and the beampatterns are shown in Figure 5a,b. The numerical results on the WPR and WSA for Example 1 and 2 are shown in Table 1.

As reported in [38], the problem of the CM-based design is that it is sensitive to stopband level constraint $\Gamma_{sb}$. If the stopband level constraint is chosen tightly, i.e., $-13$ dB, the passaband performance of the CM-based design degrades badly. This is once again proven through Figures 4a and 5a. The WPR deteriorates from 1.5541 dB to 5.8838 dB, which causes unnecessary distortion of the desired directional signal. In comparison, the proposed RCM-based method only changes the WPR from 1.5193 dB to 1.7423 dB, which can still be considered to maintain comparable passband performance and achieve better stopband attenuation. As a result, compared with the CM-based design, the proposed RCM-based design can provide better beamforming performance under different stopband constraints, demonstrating its great robustness performance.

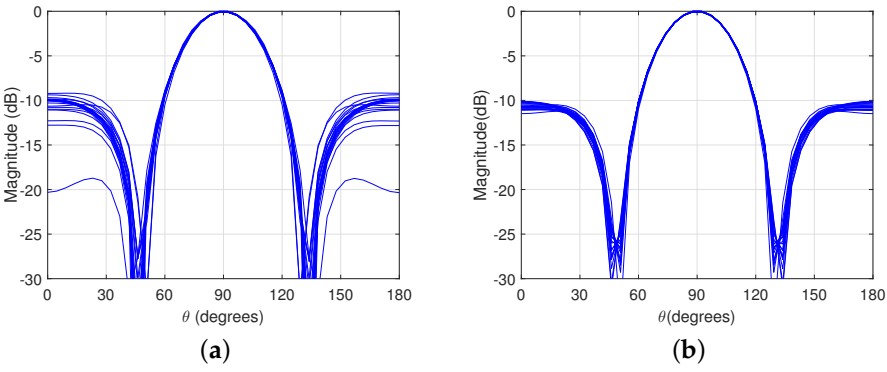

**Figure 4.** Beampatterns for the ULA, where $\Gamma_{sb}=-6$ dB. The resultant plots are drawn across 20 uniformly sampled frequencies within $[1500, 3500]$ Hz. (**a**) The CM design. (**b**) The proposed RCM design.

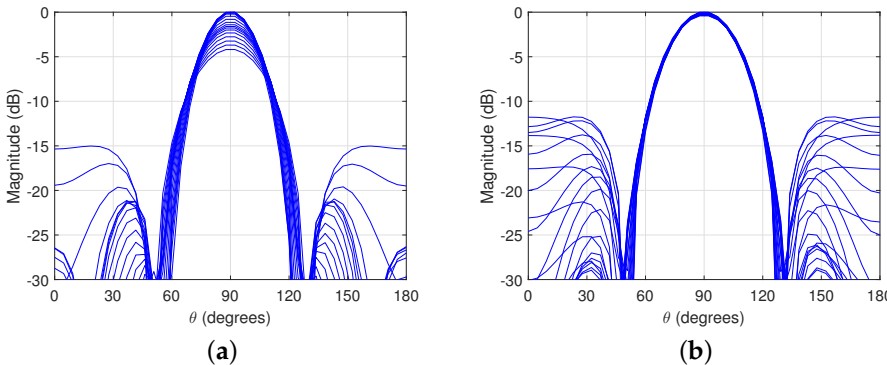

**Figure 5.** Beampatterns for the ULA, where $\Gamma_{sb}=-13$ dB. The resultant plots are drawn across 20 uniformly sampled frequencies within $[1500, 3500]$ Hz. (**a**) The CM design. (**b**) The proposed RCM design.

**Table 1.** Numerical Results of Design for Examples 1 and 2

| Approaches | $\Gamma_{sb} = -6$ dB | | $\Gamma_{sb} = -13$ dB | |
| --- | --- | --- | --- | --- |
| | $\sigma_p^{(wc)}$ (dB) | $\sigma_s^{(wc)}$ (dB) | $\sigma_p^{(wc)}$ (dB) | $\sigma_s^{(wc)}$ (dB) |
| CM | 1.5541 | 7.7327 | 5.8838 | 13.2296 |
| Proposed RCM | 1.5193 | 7.6838 | 1.7423 | 10.7635 |

*4.2. Example 3 and 4*

In this subsection, we consider the sparse array design proposed in (19) and (23). Remind that the purpose of sparse arrays is to achieve an approximate or even better beampapttern performance with fewer sensors using the same array aperture as these ULAs. To do so, we set 101 grid for potential sensor locations uniformly spread over an aperture of 0.24 cm, which is just same as the one in Example 1. $\Gamma_{pb}^{ph1}$, $\Gamma_{sb}^{ph1}$ and $\Gamma_G^{ph1}$ are calculated from the solution to the optimal weighting **w** based on the reformulated convex design in Example 1 due to (20)–(22), and we have $\Gamma_{pb}^{ph1} = 0.047$, $\Gamma_{sb}^{ph1} = 0.41$ and $\Gamma_G^{ph1} = 0.39$, respectively. The parament $\epsilon = 9 \times 10^{-5}$ and the threshold value below which sensor grids will be considered inactive is set to $\rho = 10^{-7}$.

The third example explores the impact of the number of iteration on sparsity for the robust design (23), and the number of iteration $i$ are set to 1, 4, and 8, respectively. In Figure 6, the nonzero $\|\mathbf{w}_m\|_2$ after setting the coefficients below the threshold to zeros are illustrated along the x-axis labeled as all positions of potential sensor elements. It can be seen that the $\ell_{2,1}$ sparse design with no iteration obtains the same size as the ULA, i.e., $[-0.12, 0.12]$ m, but the remaining elements are around $[-0.574, 0, 0.574]$ m, which leads to the problem that several grids remained directly adjacent, which is nearly impossible to implement these sensors in practice. Subsequently, the reweighted $\ell_{2,1}$ optimization with iterations can further sparse the array with sensor accurately located at $[-0.12, -0.0648, 0, 0.0648, 0.12]$ m respectively, overcoming the problem of adjacent girds and earn less sensor elements than the uniformly linear array.

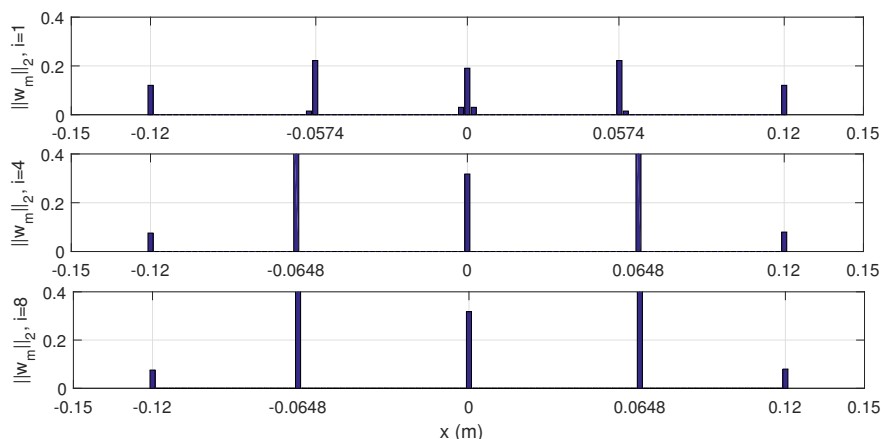

**Figure 6.** The locations of nonzero $\|\mathbf{w}_m\|_2$ along the horizontal axis.

To better understand the performance difference in the above designs, the beampatterns of different iterations and threshold value are shown in Figures 7 and 8, and parameter values are summarized in Table 2. Recall that the $\ell_{2,1}$ design has to discard coefficients below the set threshold. Hence, we also give the beampattern with large threshold value of non-iteration $\ell_{2,1}$ design in Figure 7. When choosing a larger threshold value, the main difference is that extra elements will be rounded off, especially around $[-0.574, 0, 0.574]$ m, as shown in Figure 6. And the number of remaining elements is the same as the result of multiple iterations, which is five. The large threshold value indeed spares more grids, but this method of obtaining sparsity suffers a poor passband performance of beampattern shown in Figure 8a. Results in Table 2 further confirm that the passband ripple decreases

from 1.0436 to 1.7262 dB and the stopband level decreases from 9.1358 to 7.8791 dB for $\rho$ changed from $= 10^{-7}$ to 0.01. Moreover, the robustness characteristic also has degraded for the large threshold in terms of WPR and worst-case stopband level in Table 2. Simulation results indicate that it is inappropriate for eliminating more array elements by choosing a larger threshold value, which leads to a decrease in the performance of the beampatten. On the other hand, it is also noticed that the $\ell_{2,1}$ designs with $i = 4$ and $i = 8$ show identical results on all compared results, simultaneously exhibiting optimal robustness and frequency invariant performance. This indicates that the proposed sparse design method can achieve stable optimization results with only a few iterations, and has efficient sparse beamforming performance.

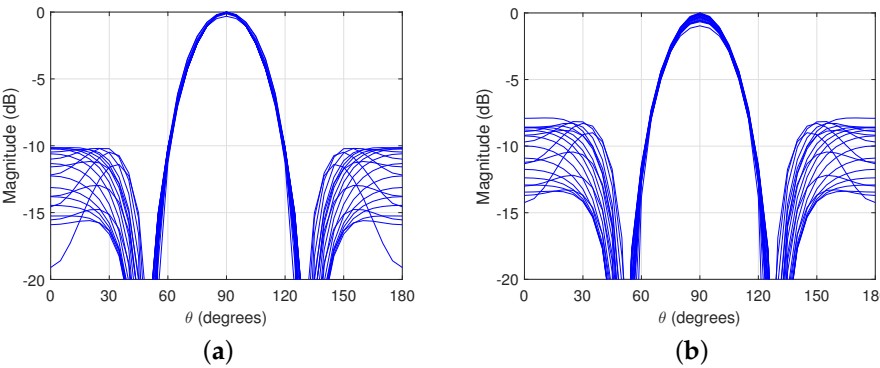

(a)                  (b)

**Figure 7.** Beampatterns for the $\ell_{2,1}$ optimization, where iteration $i = 1$. The beampattern plots are obtained across 20 uniformly sampled frequencies within the frequency band $[1500, 3500]$ Hz. (**a**) The threshold value $\rho = 10^{-7}$. (**b**) The threshold value $\rho = 0.01$.

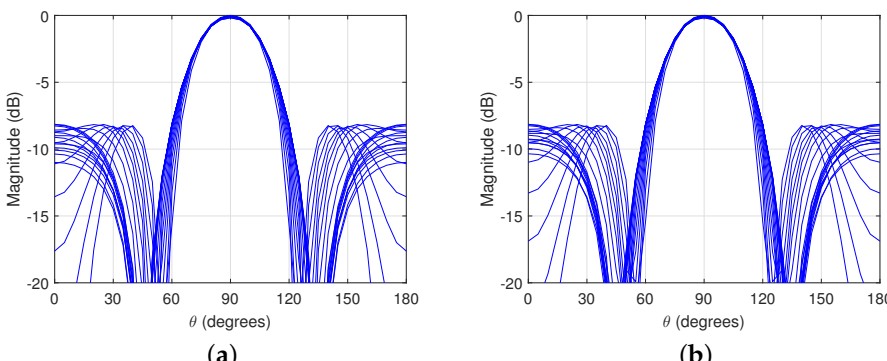

(a)                  (b)

**Figure 8.** Beampatterns for the $\ell_{2,1}$ optimization, where threshold vale $\rho = 10^{-7}$. The beampattern plots are obtained across 20 uniformly sampled frequencies within the frequency band $[1500, 3500]$ Hz. (**a**) The iteration value $i = 4$. (**b**) The iteration value $i = 8$.

Furthermore, it is curious whether the sparse array can be used for different design methods. Thus in the fourth example we choose the proposed RCM-based method, existing second-order cone programming (SOCP) based [32] and semidefinite programming (SDP) based approaches [38] to design beamformers using optimized five-element sparse array obtained from $\ell_{2,1}$ design with $i = 4$ in Example 3. The simulation data of robust broadband beamformers using various design methods with proposed five-element sparse array and seven-element ULA are recorded in Table 3. It can be seen that the WPR of all three design approaches for sparse array are slightly larger than those for ULA, while the WSA are somewhat better. Comparing the PIFs, the beamformers of the same design method have nearly the same frequency invariant properties under the two design conditions of the sensor array. Therefore, compared to scenarios using seven-element ULA under similar conditions, the proposed five-element sparse array can be applied to different WCPO design method, providing appropriate passband performance and robustness.

Table 4 shows the CPU time for the design of various beamformers, where each result is an average of twenty trials. Computations were performed on a computer with an Intel Core i7-9750H processor at 2.60 GHz and 32 GB RAM and have been implemented in Matlab R2015b. It can be found that using fewer sensors can significantly reduce the computational time in the design process of robust wideband beamformers. Based on the above discussion and analysis, the sparse design proposed in this paper can significantly reduce computational complexity and also have the advantage of reducing hardware costs in practical applications, while ensuring good passband performance and robustness of the designed broadband beamformers.

**Table 2.** Numerical Results of Design for Example 3.

| Approaches | $\sigma_p$ (dB) | $\sigma_s$ (dB) | $\sigma_p^{(wc)}$ (dB) | $\sigma_s^{(wc)}$ (dB) | PIF (dB) |
|---|---|---|---|---|---|
| Proposed RCM | 1.0202 | 9.0632 | 1.5193 | 7.6838 | −34.9116 |
| $\ell_{2,1}$ design with $i = 1$ and $\rho = 10^{-7}$ | 1.0436 | 9.1358 | 1.5194 | 8.0592 | −33.3758 |
| $\ell_{2,1}$ design with $i = 1$ and $\rho = 0.01$ | 1.7262 | 7.8791 | 2.3956 | 6.9615 | −30.0297 |
| $\ell_{2,1}$ design with $i = 4$ and $\rho = 10^{-7}$ | 0.8171 | 8.1433 | 1.3793 | 7.4694 | −34.6292 |
| $\ell_{2,1}$ design with $i = 8$ and $\rho = 10^{-7}$ | 0.8171 | 8.1433 | 1.3793 | 7.4694 | −34.6292 |

**Table 3.** Numerical Results of Design for Example4.

| Approaches | 5-Element Sparse Array | | | 7-Element ULA | | |
|---|---|---|---|---|---|---|
| | $\sigma_p^{(wc)}$ (dB) | $\sigma_s^{(wc)}$ (dB) | PIF (dB) | $\sigma_p^{(wc)}$ (dB) | $\sigma_s^{(wc)}$ (dB) | PIF (dB) |
| SOCP | 2.6462 | 9.2248 | −28.3147 | 2.6045 | 9.0276 | −28.9776 |
| SDP | 1.5315 | 8.4879 | −35.9086 | 1.4900 | 8.3515 | −34.6050 |
| Proposed RCM | 1.5332 | 8.4835 | −35.5277 | 1.5193 | 7.6838 | −34.9116 |

**Table 4.** CPU time of the Design Approaches for Example 4.

| Approaches | CPU Time(s) | |
|---|---|---|
| | 5-Element Sparse Array | 7-Element ULA |
| SOCP | 11.46 | 21.36 |
| SDP | 68.16 | 167.08 |
| Proposed RCM | 27.95 | 82.25 |

## 5. Conclusions

In this paper, we have proposed a robust sparse design approach that utilizes reweighted $l_{2,1}$ norm optimization using CM-based criterion. The significance of this method is to use sparse, i.e., fewer array elements, to achieve the same beamforming results as a ULA. To achieve this, a broadband beamformer with stable passband performance and robustness is required as an initial. Therefore, a optimal problem of CM-based design is firstly reformulated to enhance the tradeoff between beamforming performance and robustness, by separating passband respond performance from origin cost function. This reformulation allows for improved adaptability to different settings of the stopband level constraints. Furthermore, a $l_{2,1}$ norm optimization is proposed, aiming to simultaneously minimize all coefficients of the $m$th $L$-tap FIR filter for reducing the number of sensors on the predetermined grids. However, it has been observed that this results in the presence of multiple adjacent sensor locations, posing challenges in the selection of active sensors. While adopting a lenient threshold can address this issue, it may adversely affect the beamforming performance. Consequently, an iterative process with a reweighting $l_{2,1}$ term is introduced.

The proposed sparse design based on reweighting $l_{2,1}$ optimization can further improve the sparsity of sensor locations and fulfill the comparative results with fewer sensors compared to a ULA design. Meanwhile, simulation results have surprisingly shown that the optimized sparse array can be applied to various WCPO-based design methods and achieve similar results as above.

**Author Contributions:** Conceptualization, Y.B.; software, Y.B. and X.L.; validation, H.Z. and Y.J.; formal analysis, Y.B., H.Z. and Y.J.; resources, X.L.; writing—original draft preparation, Y.B.; writing—review and editing, Y.B. and Y.T.; project administration, Y.B. All authors have read and agreed to the published version of the manuscript.

**Funding:** This research was funded by the National Natural Science Foundation of China (Grant No. 62001060), the Qing Lan Project of Jiangsu Province, the Natural Science Fundation of the Jiangsu Higher Education Institutions of China (Grant No. 23KJD140001) and Changzhou Sci&Tech Program (Grant No. CJ20220256), Key Laboratory of Radar Imaging and Microwave Photonics (Nanjing University of Aeronautics and Astronautics), Ministry of Education (Grant No. NJ20220008).

**Data Availability Statement:** The data presented in this study are available on request from the corresponding author. The data are not publicly available due to follow-up studies are being further conducted.

**Conflicts of Interest:** The authors declare no conflict of interest.

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
