# Peer review of "Design of Robust Sparse Wideband Beamformers with Circular-Model Mismatches Based on Reweighted 2,1 Optimization"

_remotesensing, doi:10.3390/rs15194791_

Round 1

Reviewer 1 Report

The paper proposes a robust sparse wideband beamformers design approach based on reweighted â„“21 norm optimization with circular-model mismatches variation. Simulation experiments show that the reformulated CM-based algorithm effectively improves the performance of beamformer. Moreover, the sparse iterative method can apply less number of sensors to achieve similar results. This has potential to reduce computational complexity and hardware costs in practical applications. 

The topic is interesting and this paper is well-written, the results are encouraging. 

I recommend that the paper be accepted for publication, provided that following condition is satisfied:

1) The specific locations of the sparse sensors are suggested to be marked in Figure 5 or illustrated in Section IV.

2) A larger size of Figure 1 is expected.

3) After Equation 7, please add the definition of delta_a, delta_phi and delat_d. 

4) In Section 2.2, it seems that the Equation 10 can be omitted and the description can be adjusted accordingly. 

5) Correct the writing errors. Such as: in abstract “wideband beamformers has….”; in line 143 the reference [?]. 

6) Update references from the past five years

7) Please check the title of reference 22 in line 345;

N/A

Reviewer 2 Report

The paper discusses design of Robust Sparse Wideband Beamformers with Circular-Model Mismatches based on Reweighted `2,1 Optimization.

The following are suggestions towards improvements in the paper:

1. The paper presents literature review. The gaps is to be identified in the review which could lead to contributions of the paper.

2. The methodology of the paper could be detailed in the form of work flow diagram.

3. results could be depicted in the form of tables and graphs.

4. some analysis and discussion could be added on as observations and inferences based on the results.

5. This could lead to logical conclusions

Reviewer 3 Report

This paper  proposed a robust sparse wideband beamformers with circular-model mismatches based on reweighted l2,1  optimization, this is an classical problem, the  authors use  the reweighted sparse  reweighted  to  solve this problem.  Some comments are given as follows:

1.     How is the Taylor expansion in formula 7 implemented? It is recommended to provide details.

2.     In line 143, the reference is missing.

3.     What is the meaning of Bd  in  the equation 10.

4.     Suggest comparing the complexity of the proposed algorithm with existing algorithms.

5.     The performance improvement of the proposed method is not evident from Figure 4. Please provide an explanation.

6.     There have been few references in recent years, and it is recommended to supplement them.

Minor editing of English language required

Round 2

Reviewer 3 Report

All question I raised have been addressed,I  think this paper can be accepted at present form for  publication.